# Low pH Stress Enhances Gluconic Acid Accumulation with Enzymatic Hydrolysate as Feedstock Using *Gluconobacter oxydans*

**Lin Dai [1], Zhina Lian [1,\*], Yixiu Fu [2], Xin Zhou [1,\*], Yong Xu [1], Xuelian Zhou [3], Boris N. Kuznetsov [4] and Kankan Jiang [2,\*]**

[1] Jiangsu Co-Innovation Center of Efficient Processing and Utilization of Forest Resources, College of Chemical Engineering, Nanjing Forestry University, Nanjing 210037, China

[2] School of Basic Medical Sciences and Forensic Medicine, Hangzhou Medical College, Hangzhou 310059, China

[3] Jiangsu Province Key Laboratory of Biomass Energy and Materials, Institute of Chemical Industry of Forest Products, Chinese Academy of Forestry, Nanjing 210042, China

[4] Institute of Chemistry and Chemical Technology SB RAS, FRC KSC SB RAS, Siberian Federal University, Krasnoyarsk 660041, Russia

\* Correspondence: lianzhina@njfu.edu.cn (Z.L.); xinzhou@njfu.edu.cn (X.Z.); jiangkankan@126.com (K.J.)

**Abstract:** Gluconic acid has been increasingly in demand in recent years due to the wide applications in the food, healthcare and construction industries. Plant-derived biomass is rich in biopolymers that comprise glucose as the monomeric unit, which provide abundant feedstock for gluconic acid production. *Gluconobacter oxydans* can rapidly and incompletely oxidize glucose to gluconic acid and it is regarded as ideal industrial microorganism. Once glucose is depleted, the gluconic acid will be further bio-oxidized to 2-ketogluconic acid by *Gluconobacter oxydans*. The endpoint is difficult to be controlled, especially in an industrial fermentation process. In this study, it was found that the low pH environment (2.5~3.5) could limit the further metabolism of gluconic acid and that it resulted in a yield over 95%. Therefore, the low pH stress strategy for efficiently producing gluconic acid from biomass-derived glucose was put forward and investigated with enzymatic hydrolysate. As a result, 98.8 g/L gluconic acid with a yield of 96% could be obtained from concentrated corncob enzymatic hydrolysate that initially contained 100 g/L glucose with 1.4 g/L cells loading of *Gluconobacter oxydans*. In addition, the low pH stress strategy could effectively control end-point and decrease the risk of microbial contamination. Overall, this strategy provides a potential for industrial gluconic acid production from lignocellulosic materials.

**Keywords:** *Gluconobacter oxydans* (*G. oxydans*); low pH; gluconic acid; glucose; enzymatic hydrolysate

## 1. Introduction

Lignocellulosic biomass, as the globally most abundant renewable resources, is comprised primarily of cellulose, hemicellulose and lignin, and exhibits great potentials in many applications in today's age of sustainable-development awareness [1]. Glucose-based cellulose, which represents a large portion of lignocellulose, is commonly refined to produce bioethanol and other chemicals. However, the high cost caused by biorefinery processes compels the need to develop value added products [2]. Gluconic acid (GA) is a kind of polyhydroxycarboxylic acid, and its basic salts, such as calcium gluconate and sodium gluconate, play essential and diverse roles in medical, food, and construction industries [3]. It is worth mentioning that sodium gluconate is the most frequently used superplasticizer and that it exhibits good application in dispersing cement/concrete particles and hence increases the fluidity of concrete-based materials [4]. In addition, the purity requirement

for GA as a cement/concrete additive is low; therefore, the GA that derives from enzymatic hydrolysate and bio-oxidation need not undergo further separation and purification processes, and can be directly used as a retarder.

Currently, various approaches for producing GA or gluconate salt, including chemical, electrochemical, and biochemical have been reported. Herein, GA could be chemically produced by oxidation of glucose with hypochlorite solution or in the presence of a bromide [5]. Although chemical conversion involves a single step, it limits selectivity towards GA. Besides this, environmental toxicity and biological hazards also make it unfavorable. Comparatively, the bioprocess can reduce side reactions during chemical production, making it an economical and attractive method for GA production. Common biological methods for GA production are to use free glucose oxidase (GOD) and GOD-containing microorganisms or glucose dehydrogenase (GDH)-containing microorganisms [6]. Nevertheless, using GOD for producing GA is not typically considered as the ideal method because of its highly stability-dependent feature and low production [7]. Microbial fermentation provides a competitive advantage over pure enzyme catalysis for its low cost and process stability [8]. The previous literature shows a consensus that submerged fermentation employing *Aspergillus niger* is the most viable industrial production method [9–11]. The synthesis of GA by *Aspergillus niger* mainly depends on GOD and catalase around its cell wall; however, its slow growth and acid intolerance lead to an unsatisfactory yield [8,12,13].

*Gluconobacter oxydans* (*G. oxydans*) has also been reported to be able to produce GA, and it catalyzes glucose to GA using a different enzyme from *Aspergillus Niger*; *G. oxydans* was regarded as the preferable microorganism due to a better tolerance for lignocellulose degraded inhibitors and high content of sugars [14]. *G. oxydans* is a gram-negative bacterium belonging to the *Acetobacter* genus, which is strictly aerobic and known for its ability to incompletely and rapidly oxidize a variety of sugars and alcohols, such as xylonic acid and GA [15]. *G. oxydans* can be uncoupled from fruits and flowers, and has been proved to be innocuous or no infection risk for healthy people, it therefore can be used in large-scale industrial production [16]. Equipped with a series of membrane-bound aldehyde dehydrogenase and alcohol dehydrogenase, *G. oxydans* can catalyze a variety of substrates; so far, the reported industrialized products are aldonic acid, erythrulose and dihydroxyacetone [17,18]. The glucose metabolic pathway of *G. oxydans* is mainly oxidation to GA by membrane-bound GDH, and the GA will be further converted to 2-ketogluconic acid (2-KGA) by membrane-bound gluconate dehydrogenase (GADH), while partial glucose is carried into the cell by transporters and phosphorylated into Entner-Doudoroff pathway (ED) or pentose phosphate pathway (PPP) [19,20]. The literatures reveal that this strain has the potential to be an industrial microorganism to produce GA [21–23]. Nevertheless, the main problem to be solved urgently is that the product of GA will be further oxidized to 2-KGA, which will reduce the yield of GA [19].

Within the context of the ever-increasing demand for GA and its derivatives, exploring low-cost and environmental-friendly feedstocks is of pronounced interest [24]. Zhang et al. successfully obtained GA from corn stover hydrolysate with the yield of 87.6% by *G. oxydans* [25]. Zhou et al. achieved a high yield of 93.9% GA from dilute acid pretreated by corn stover with a regulator ($ZnCl_2$) in whole-cell catalysis of *G. oxydans* [24]. With a rationalized understanding of some representative cases, in this study, a more feasible method through utilizing lignocellulosic glucose to produce GA directionally was tested. Firstly, different pH neutralizers were used for maintaining acidic fermentation process and bio-catalytic respects were compared. Simultaneously, the growth of *G. oxydans* and the performance of fermentation in different pH conditions were investigated, and the enzymatic activities were determined for further validation. The focus of this study lay in the regulation of pH on *G. oxydans* glucose metabolism, and it was found that low pH condition could effectively limit the further oxidization of GA. Eventually, GA with higher yield was achieved using lignocellulosic hydrolysate as feedstock with no neutralizer addition; moreover, the low pH stress strategy could effectively control the end-point without byproducts and can potentially decrease the risk of microbial contamination by

the acidic environment, which provides a promising strategy for green production of high-value chemicals.

## 2. Materials and Methods

### 2.1. Bacterial Strain and Culture Conditions

*G. oxydans* was derived from *G. oxydans* 621H and maintained on 1.5% agar plate with 50 g/L sorbitol and 5 g/L yeast extract at 4 °C in Nanjing Forestry University [26]. Cells were pre-cultivated in the culture consisting of 50 g/L sorbitol and 5 g/L yeast extract in a 250 mL baffled Erlenmeyer flask at 30 °C for 24 h, with constant shaking at 220 rpm. Activated cells were collected and transferred to fresh growth medium (500 mL) in a 2 L baffled Erlenmeyer flask with 100 g/L sorbitol and 10 g/L yeast extract. Bacteria cells were then incubated for 16 h at 30 °C and 220 rpm. After incubation, cells of *G. oxydans* were recovered by refrigerated centrifugation (Avanti J-26 XP, Beckman Coulter) for 5 min at 8000 rpm. Finally, cell pellets were collected and loaded into the fermentation system as required (the initial $OD_{600}$ = 2).

Fermentation was performed in a medium consisting of yeast extract (5 g/L), $(NH_4)_2SO_4$ (5 g/L), $KH_2PO_4$ (2 g/L), $K_2HPO_4$ (1 g/L), $MgSO_4$ (0.5 g/L), and enzymatic hydrolysate in baffled Erlenmeyer flasks or 1 L stirred-bioreactor (BIOTECH-1JD-7000A) [27]. The fermentation assays were conducted at 30 °C and 220 rpm for 36 h in flasks or 30 °C, 1.5 vvm and 500 rpm in 3 L bioreactor. An air compressor was used for supplying air in bioreactor. In addition, calcium carbonate ($CaCO_3$), 10% $NH_3 \cdot H_2O$ and 50% NaOH solutions were used to control the pH of the fermentation broth. Here, 2 g $CaCO_3$ was directly pre-loaded into the medium before fermentation; 10% $NH_3 \cdot H_2O$ and 50% NaOH solutions were manually loaded into the flasks at hourly intervals during fermentation and the pH was every time adjusted to 5.5.

### 2.2. Enzymatic Hydrolysis of Pretreated Corncobs

Corncobs were used as lignocellulosic feedstocks in this study which contained 34.2% cellulose, 32.8% hemicellulose, and 19.2% lignin. Corncobs were first pretreated with 1% (*w/w*) $H_2SO_4$ at 150 °C for 30 min, with a solid/liquid ratio of 1:10. A 10 L reaction volume was autoclaved with a loaded reaction and the mixture centrifuged to separate solid residue before enzymatic hydrolysis. For enzymatic hydrolysis, pretreated corncob solids were washed by water to obtain a neutral pH solid and air-dried for 24 h. Enzymatic hydrolysis was then performed in a 3 L bioreactor for 24 h at 50 °C and 150 rpm metal axis stirring, and the solids loading was 10% (*w/w*) [28]. The pH was maintained at 4.8 by online adding 50% NaOH solution during enzymatic hydrolysis. The amount of enzyme used was 20 FPIU/g-glucan cellulase (C2730, Celluclast® 1.5 L, Novozymes, Sigma Co., Shanghai, China). After the process of enzymatic hydrolysis, the supernatant was collected by centrifuged at 5000 *g* for 5 min. The enzymatic hydrolysate with a glucose content of 56.6 g/L was then diluted or concentrated to the required concentration (50–200 g/L) by adding distilled waste or rotary vacuum evaporation (R-200, BÜCHI), and the initial pH of enzymatic hydrolysate after diluted or concentrated was adjusted to 6.5 by adding NaOH.

### 2.3. Determination of the Enzymatic Activity

Dehydrogenases on *G. oxydans* cell membrane were determined using a microplate reader with 2,6-dichlorophenol indophenol (DCPIP) as electron acceptor [29,30]. In this reaction, the microtiter plate with 200 µL reaction system orbital oscillated at 218.3 rpm for 1 s at 10 s intervals for 30 cycles and the absorbance were measured at 600 nm. Formula (1) was used to calculate the enzyme activities:

$$U = \frac{V_t \frac{\Delta A}{\Delta t}}{V_s l \varepsilon} \tag{1}$$

In the formula, $U$ was the enzyme activity; $V_t$ (mL) was the total reaction volume; $\frac{\Delta A}{\Delta t}$ represented the slope of absorbance plotted against time; $V_s$ (mL) was the volume of bacterial liquid; $l$ (cm) was the diameter of microtiter plate well; and $\varepsilon$ (L·(mmol·cm)$^{-1}$) was the molar extinction coefficient of DCPIP (the slope of the standard curve of DCPIP solution).

*2.4. Analytical Methods*

The chemical composition of the corncob solids and dilute acid-pretreated corncob solids were determined according to the protocol from the National Renewable Energy Laboratory. Briefly, 20–80 meshes corncobs were first hydrolyzed by 72% (*w/w*) $H_2SO_4$ at 30 °C for 1 h; then, the 72% (*w/w*) $H_2SO_4$ was diluted to 4% (*w/w*) by loading distilled water. Finally, the sealed bottle was autoclaved at 121 °C for 1 h. After autoclave, the suspension slurry was used for analysis carbohydrates and acid-soluble lignin.

The optical density (OD) of *G. oxydans* was determined using a UV-vis spectrophotometer (Spectrumlab 752 s) at 600 nm. Cell concentration was calculated using the following Formula (2) [31]:

$$C_{cell} = 0.68\ OD_{600} \tag{2}$$

Initial glucose concentration in the enzymatic hydrolysate was determined using high performance liquid chromatography (HPLC, Agilent 1260) [32]. A Dionex ICS-3000 high-performance anion exchange chromatography was employed to detect glucose, GA and 2-KGA content during the fermentation process [33].

GA yield was calculated according to the following Formula (3):

$$GA\,yield(\%) = \frac{GA\,concentration}{initial\,glucose\,concentration \times 1.089} \times 100 \tag{3}$$

$$2-KGA\ yield\ (\%) = \frac{2-KGA\ concentration}{initial\ glucose\ concentration \times 1.078} \times 100 \tag{4}$$

**3. Results and Discussion**

*3.1. Effects of Different Neutralizers on Glucose Metabolism of G. oxydans*

*G. oxydans* is considered an ideal strain for industrial production of sugar acids due to its ability to incompletely oxidize various substrates with improved bio-catalytic efficiency [34]. *G. oxydans* can oxidize sugars to some acidic products, which will decrease the pH of the medium during fermentation [35]. It is known that *G. oxydans* oxidizes glucose to GA, as the intermediate, which will be further oxidized to 2-KGA, and the process is dependent on membrane-bound dehydrogenases [36]. Moreover, membrane-bound dehydrogenases of *G. oxydans* are sensitive to pH conditions [37]. Therefore, pH is regarded as an important factor, affecting fermentation of glucose by *G. oxydans*. According to the existing literature, the optimum pH for GDH is 5~6 [38]. The increasing accumulation of GA will rapidly decrease the pH of broth to 2.5, at this point, the bound membrane enzymes that work for energy generation by dehydrogenation reaction are almost completely inactivated [38]. Overall, pH neutralizers are required to be loaded for maintaining a favorable pH environment during glucose oxidation process [39].

In this study, $CaCO_3$, NaOH and $NH_3 \cdot H_2O$ as the most frequent neutralizers were used to maintain pH at 5~6. The effects of these neutralizers on fermentation performance were investigated and the data was displayed in the Figure 1. In the fermentation process, $CaCO_3$ powder was pre-added in the flasks before fermentation, and 50% NaOH solution and 10% $NH_3 \cdot H_2O$ solution were manually added into the flasks every hour during fermentation, while the pH was adjusted to 5.5 until the glucose was completely consumed. Figure 1 shows that 36.1 g/L, 32.5 g/L and 27.7 g/L of GA were obtained at 3 h with $CaCO_3$, NaOH and $NH_3 \cdot H_2O$, respectively; and the corresponding average volumetric productivity was 12.0, 10.8 and 9.2 g/L/h, respectively. In addition, it was also found that the average volumetric productivity improved with increasing fermentation time. Finally, 100 g/L glucose could be used up at 6 h, except the assay using $NH_3 \cdot H_2O$ as neutralizer.

Overall, *G. oxydans* could rapidly bio-oxidize glucose into GA with pH maintained at 5~6, and the difference with different neutralizers was not significant.

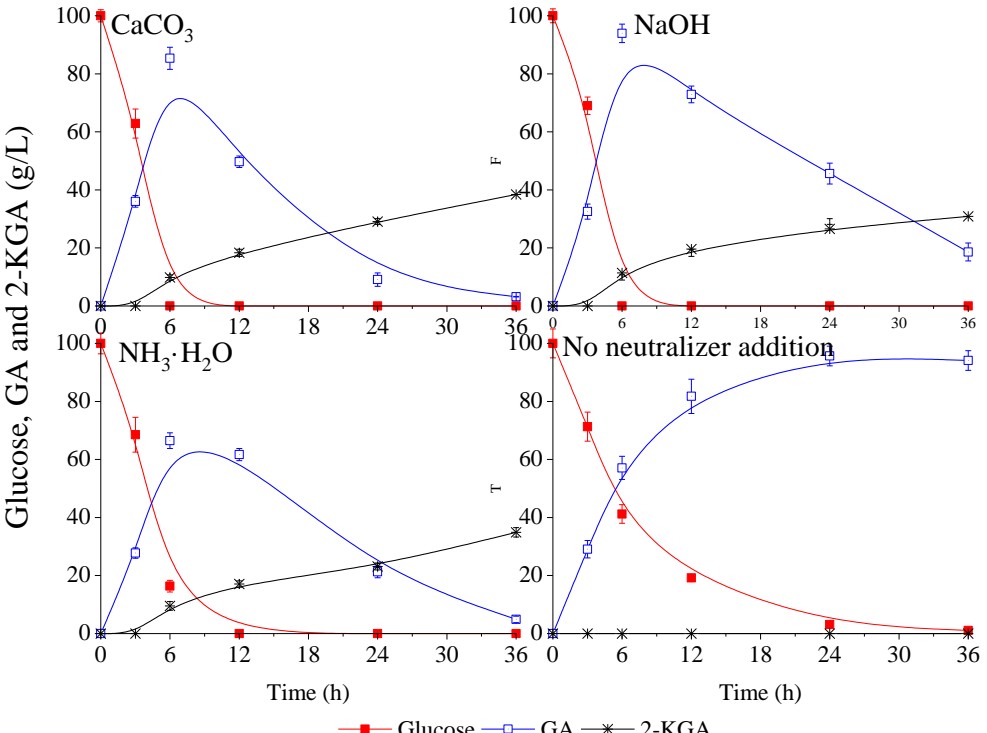

**Figure 1.** Effect of pH neutralizers on the glucose metabolism kinetics of *Gluconobacter oxydans* (GA: gluconic acid; 2-KGA: 2-ketogluconic acid). Three different prevalent neutralizers were chosen: CaCO₃, NaOH, NH₃·H₂O. The blank control was no neutralizer addition (none).

As shown in Figure 1, the majority of GA was further metabolized when the pH was maintained at 5~6. It was found that the GA was not only converted into 2-KGA (yields with different pH neutralizers were all lower than 40%), but also oxidized and decomposed. In addition, the kinetic curves of 2-KGA with different neutralizers indicated little difference among the three experiments at a random time, suggesting that the pH neutralizer was not the main influence factor for glucose metabolism of *G. oxydans*.

As a control, the assay without pH maintenance in the process of fermentation (equivalent to always being kept in a low pH environment) was also evaluated, and the results were also showed in the Figure 1. It could be observed that approximately 20 g/L glucose remained in the medium at 12 h. Although the productivity was slowed down, almost quantitative conversion of glucose to GA was produced, and the glucose was almost used up at 24 h, while 96.4 g/L GA was accumulated in the broth. It seemed that pH control was not necessary for the GA fermentation when the initial glucose concentration was 100 g/L, and no neutralizer addition was benefit for subsequent isolation and purification. The results suggest that GA production by *G. oxydans* without pH control is feasible under the appropriate glucose concentration.

Although the metabolic rate of glucose somewhat declined, it was interesting to find that the generated GA was not further metabolized even though the glucose was completely consumed and almost no 2-KGA accumulated until 36 h. Taken together, these results suggested that low pH limited glucose metabolism and the production of 2-KGA was more strongly inhibited. The catalytic reaction endpoint without undesirable byproduct 2-KGA could be further exploited by controlling the pH of fermentation under the precondition with no influence on bacterial survival.

### 3.2. The Effects of Different pH Environments on Glucose Metabolism of G. oxydans

　　To examine the effects of the pH environment on the performance of *G. oxydans* for glucose metabolism, fermentation experiments with different pH environments (pH 2.5, 3.5, 4.5, 5.5, and 6.5) were performed in a 1 L bioreactor. Membrane-bound *G. oxydans* GDH is known to oxidize glucose to GA and most of the GA will be further oxidized to 2-KGA by the membrane-bound GADH. Therefore, influence of pH on *G. oxydans* glucose metabolism can be explained by the catalytic activity of GDH and GADH. Figure 2 showed glucose metabolism, bacterial growth kinetics and enzyme activities (GDH and GADH) under different pH environments. It was showed that the bioreactor could maintain a relatively stable pH by online NaOH auto-loading. As shown in Figure 2a, 100 g/L glucose was depleted at 6 h with a GA yield of 77.0%, while 5.2 g/L 2-KGA was produced in a 2.5 pH environment. However, at 12 h, only 5.8 g/L 2-KGA was accumulated, implying that further reaction time has no significant impact on 2-KGA increment. The production and average productivity of GA and 2-KGA under different pH environments are summarized in Table 1.

**Table 1.** Gluconic acid and 2-ketogluconic acid production by *Gluconobacter oxydans* in different pH environment.

| pH | Production (G/L) | | Average Productivity (G/L/h) | |
|---|---|---|---|---|
| | 3 H (GA) | 12 H (2-KGA) | 0–3 H (GA) | 9–12 H (2-KGA) |
| 2.5 | 72.7 | 5.8 | 24.2 | 0.08 |
| 3.5 | 79.6 | 12.1 | 26.5 | 0.69 |
| 4.5 | 87.3 | 28.9 | 29.1 | 3.33 |
| 5.5 | 90.3 | 48.6 | 30.1 | 6.20 |
| 6.5 | 69.0 | 13.2 | 23.0 | 2.12 |

　　As these results demonstrated, lower pH (pH = 2.5 and pH = 3.5) was associated with slight decrease in GA production, whereas higher pH environments (pH = 4.5 and pH = 5.5) resulted in average GA productivity of 29.1 g/L/h and 30.1 g/L/h at 3 h, respectively. Obviously, the productivities were significantly enhanced in bioreactor compared with the experiments that conduced in flasks. *G. oxydans*, as an obligate aerobic bacterium, relies strongly on the oxygen [40]. In a shake flask, the available oxygen is mainly from surface aeration; in bioreactor the oxygen is supplied by compressed air, and vigorous stirring will result in a good oxygen transfer. Zhou et al., proved that high aeration and agitation in bioreactor could efficiently improve oxygen transfer rate and then boost productivity performance for bio-oxidizing xylose into xylonic acid with *G. oxydans* [41]. Therefore, it was inferred that the great improvement was mainly caused by the better oxygen supply and oxygen transfer rate.

　　Remarkably, both GA and 2-KGA production declined under the pH 6.5 environment, suggesting that pH 6.5 was not ideal condition for glucose and GA metabolism. The results in Table 1 showed that 2-KGA average productivity was much lower in pH 2.5 and 3.5 environments, with only 0.08 g/L/h produced at 9–12 h in pH 2.5 condition. In contrast, 3.33 g/L/h of 2-KGA was obtained in pH 4.5 environment. At 12 h, 48.6 g/L 2-KGA was accumulated with an average productivity of 6.2 g/L/h under pH 5.5 condition. This was the maximum productivity of 2-KGA in this study. On the basis of glucose metabolism, lower pH conditions (<3.5) strongly inhibited 2-KGA production by *G. oxydans* but only slightly impacted GA production.

　　*G. oxydans* grew by glucose fermentation, and the bacterial growth curves in Figure 2a showed that *G. oxydans* grew well in different pH environments when 100 g/L glucose was provided as a carbon source. Lower pH (2.5 and 3.5) did not affect *G. oxydans* growth as the OD increased from 2.0 to about 4.4. Consistent with this finding, 26.5 g/L glucose remained at 3 h under pH 2.5 conditions but GA composition was 72.7 g/L, equivalent to 88% of the theoretical yield. The lower GA yields found in pH 2.5 and 3.5 environments did not match glucose consumption, suggesting that *G. oxydans* uses more glucose for proliferation to

resist unfavorable environments. Additionally, when the pH of the fermentation broth was 6.5, cells grew at a slightly lower rate than when grown in the other four pH environments. Overall, pH did not significantly influence *G. oxydans* growth but might affect the glucose-related metabolism.

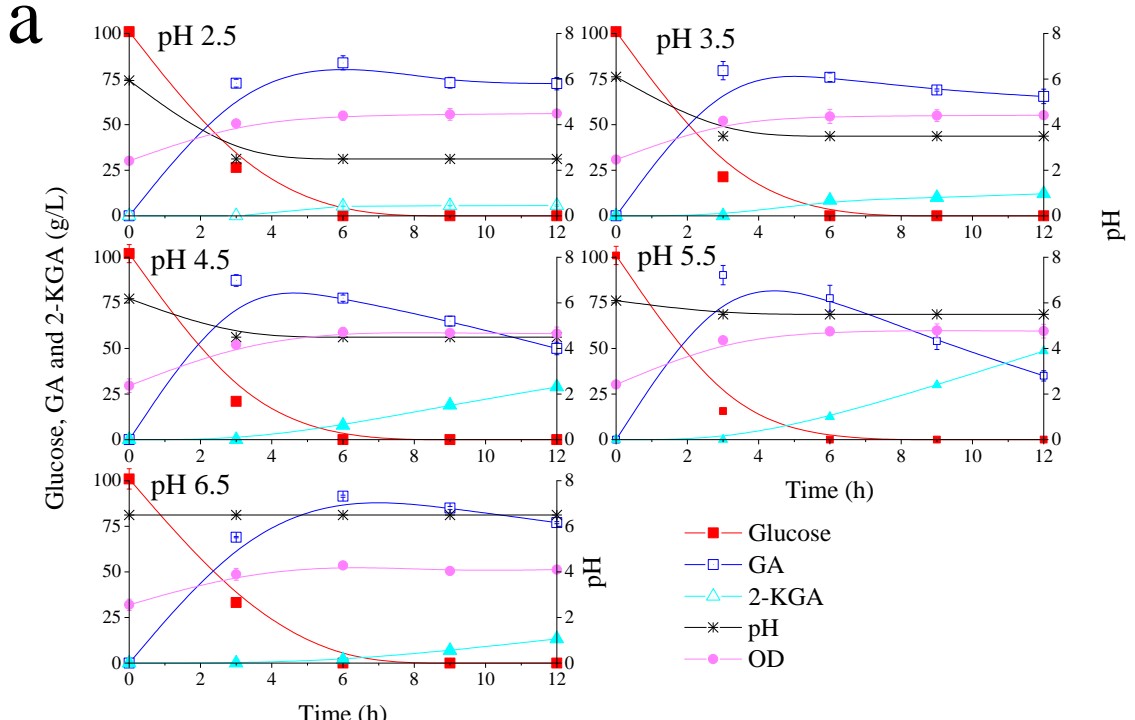

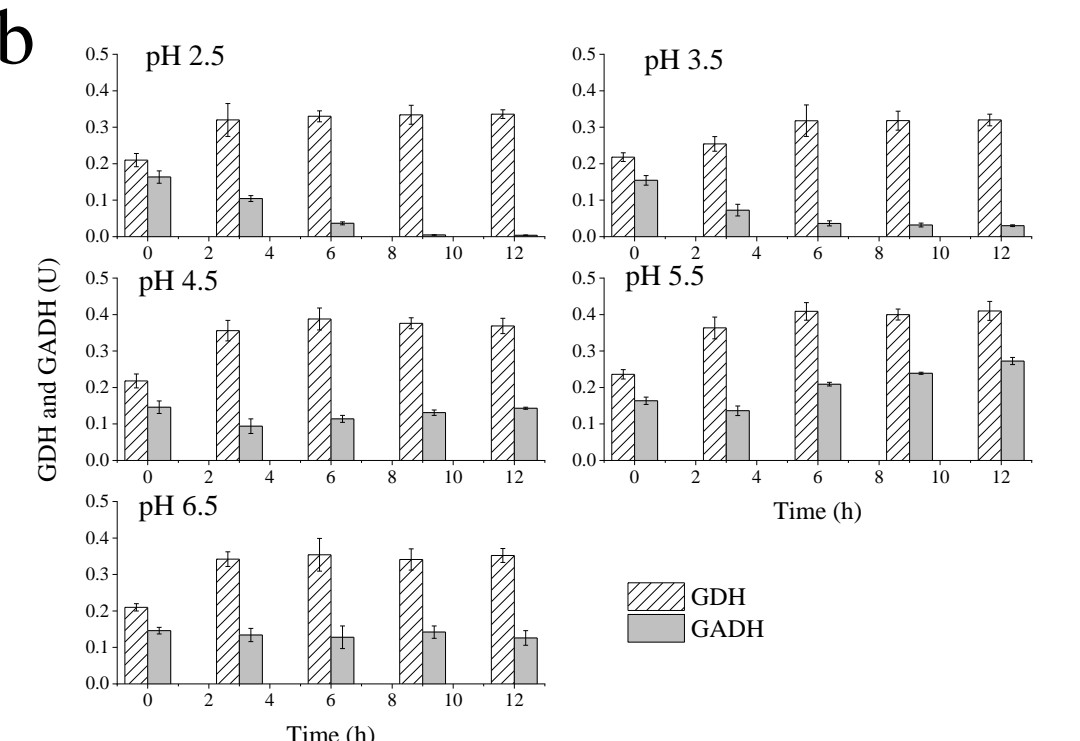

**Figure 2.** *Gluconobacter oxydans* growth kinetics, glucose metabolism kinetics (GA: gluconic acid; 2-KGA: 2-ketogluconic acid) on cell membrane (**a**); GDH (glucose dehydrogenase) and GADH (gluconic acid dehydrogenase) enzyme activities (**b**) at pH 2.5–6.5.

To further explain the discrepancy in glucose and GA metabolism under different pH environments, catalytic activities of membrane-bound GDH and GADH were measured. Figure 2b shows *G. oxydans* GDH and GADH activities at different times under pH 2.5, 3.5, 4.5, 5.5 and 6.5 environments, respectively. The results in Figure 2b illustrated that enzyme activities were similar at 0 h under different pH conditions. This was because initial bacteria concentration was the same for each determination. The results showed that GDH activities increased at 3 h or 6 h of fermentation, and then plateaued across all pH environments. This period happened to be the bacteria log phase of growth, suggesting that GDH activity could be enhanced when *G. oxydans* grew faster. Consistent with glucose metabolism kinetics, GDH activity at pH 4.5 and 5.5 was higher than other pH environments. Comparing all five pH conditions, GADH activity was the highest in a pH 5.5 environment. This explained highest 2-KGA production obtained under this pH condition. Interestingly, the activity of GADH in lower pH environments was substantially inhibited, especially in a pH 2.5 environment, which implied that the accumulation of 2-KGA could be controlled by the low pH environment. Taken together, these results suggested that pH 5.5 was the most favorable environment for *G. oxydans* glucose and GA oxidation, while low pH (2.5–3.5) inhibited 2-KGA production.

### 3.3. Direct Production of GA from Lignocellulosic Hydrolysate in Low pH Environment

Lignocellulosic biomasses contain a large amount of cellulose, which can be enzymatic hydrolyzed into glucose after pretreatment; subsequently, the free cellulosic-glucose can be oxidized to GA. Thus, lignocellulosic biomass as substrate for sustainable production of GA is also environmentally friendly, renewable, and does not compete with limited global food supplies [42]. Corncob, a common agricultural waste, contains 32–45% cellulose that can be enzymatically hydrolyzed to fermentable sugars [43,44]. In this study, diluted acid pretreated corncob solid was used to produce glucose by cellulose hydrolysis and the enzymatic hydrolysate was then subjected to *G. oxydans* fermentation. Glucose metabolism kinetics were compared for the enzymatic hydrolysate under the two pH environments ($CaCO_3$ added and no neutralizer addition) to validate a previous conclusion that low pH environment inhibited 2-KGA production. Figure 3 shows that glucose was almost depleted at 12 h, meanwhile, 70.4 g/L GA and 9.9 g/L 2-KGA were produced in the case of $CaCO_3$ as neutralizer. Results of multiple experiments indicated that when glucose was depleted or neared depletion, GA would inevitably oxidize to 2-KGA. However, in the low pH experiment (pH was not adjusted), glucose was almost depleted after 24 h and 90.1 g/L GA was obtained with a yield of 90.9%. Notably, almost no 2-KGA was collected.

Because lignocellulose is a rich source of high-value chemicals, exploiting this abundantly available renewable biomass is commercially sustainable. In this study, to determine optimal enzymatic hydrolysate concentration for *G. oxydans* oxidation and identify precise low pH conditions in which 2-KGA production is inhibited, different enzymatic hydrolysate concentrations were used to produce GA. Figure 4 shows the fermentation performance of *G. oxydans* in corncob enzymatic hydrolysate that contained 50 g/L, 100 g/L, 150 g/L and 200 g/L glucose, respectively, and pH neutralizer was unused for fermentation. First, the results showed that 50 g/L glucose could be depleted after 12 h fermentation, yielding 53.5 g/L GA, and only 2 g/L 2-KGA was generated at the end. A comparable fermentation performance of 96% GA yield at 36 h was obtained with enzymatic hydrolysate containing 100 g/L glucose as feedstock.

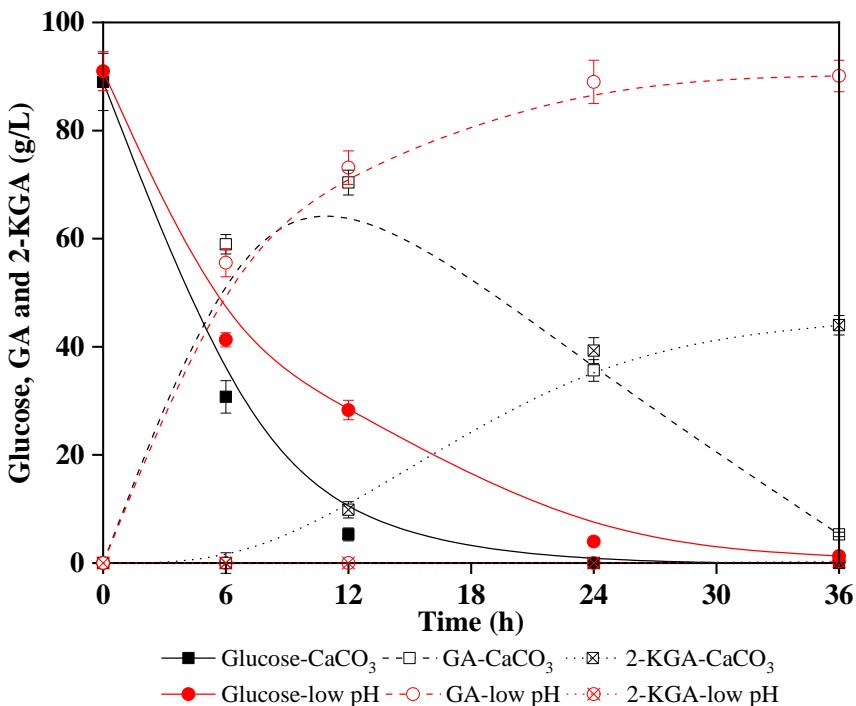

**Figure 3.** Comparison of *Gluconobacter oxydans* performance in GA (gluconic acid) and 2-KGA (2-ketogluconic acid) production from corncob hydrolysates under a low pH environment and a slightly acidic pH environment.

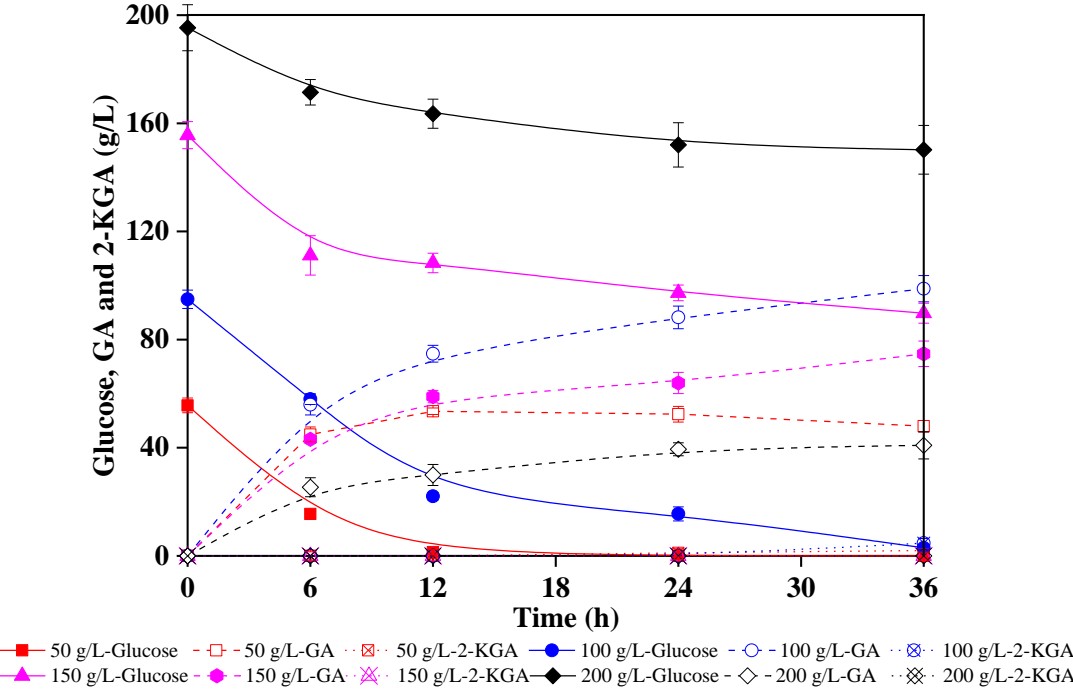

**Figure 4.** Comparison of GA (gluconic acid) and 2-KGA (2-ketogluconic acid) production performance of *Gluconobacter oxydans* in corncob enzymatic hydrolysates with different concentrations.

However, the glucose catalysis ability of *G. oxydans* declined significantly in 150 g/L and 200 g/L corncob enzymatic hydrolysate, yielding only 74.7 g/L and 40.9 g/L GA at 36 h, respectively. This might be caused by the higher titer of salt in concentrated hydrolysate from the hydrolysate preparation process. In addition, the higher concentration enzymatic hydrolysate with higher viscosity negatively impacted the oxygen transfer. In

general, hydrolysates with glucose content of more than 100 g/L were not suitable for GA production by *G. oxydans*. However, this was not a problem since the highest concentration of industrial enzymatic hydrolysate is 100 g/L. This study successfully provides a feasible strategy for producing GA from corncob enzymatic hydrolysate and effectively controlling the fermentation endpoint by maintaining a low pH environment. In addition, almost no by-products were generated at the end of the fermentation, offering significant implications for industrial production of high-value sugar acids from lignocellulosic biomass.

## 4. Conclusions

In this study, the effects of pH environments on GA fermentation by *G. oxydans* were investigated and all results indicated that the low pH environment was conducive to inhibit 2-KGA by-product generation, improve GA production, and control the endpoint of the fermentation process. As a result, 96% GA production could be achieved using concentrated enzymatic hydrolysate that contained 100 g/L glucose as feedstock, with almost no 2-KGA was generated by the low pH stress strategy, which enabled industrial GA production from lignocellulosic materials.

**Author Contributions:** Conceptualization, K.J. and X.Z. (Xin Zhou); data curation, Z.L.; formal analysis, L.D.; methodology, X.Z. (Xuelian Zhou); investigation, L.D.; resources, Z.L.; software, Y.F.; supervision, X.Z. (Xin Zhou); validation, Z.L.; visualization, K.J.; writing—original draft, L.D.; writing—review and editing, B.N.K. and Y.X.; All authors have read and agreed to the published version of the manuscript.

**Funding:** This research was funded by the financial support from the National Natural Science Foundation of China (31901270) and the Postgraduate Research & Practice Innovation Program of Jiangsu Province (SJCX22_0324), Qing Lan Project of Jiangsu Province, China.

**Institutional Review Board Statement:** Not applicable.

**Informed Consent Statement:** Not applicable.

**Data Availability Statement:** Not applicable.

**Acknowledgments:** The authors acknowledge the Advanced Analysis and Testing Center of Nanjing Forestry University.

**Conflicts of Interest:** The authors declare no conflict of interest.

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
