# Peer review of "Low pH Stress Enhances Gluconic Acid Accumulation with Enzymatic Hydrolysate as Feedstock Using Gluconobacter oxydans"

_fermentation, doi:10.3390/fermentation9030278_

Round 1
Reviewer 1 Report
Gluconic acid is useful substance produced with Gluconobacter spp. from glucose. However, Gluconobacter spp. oxidize gluconic acid further to 2-ketogluconic acid, which reduces titer and yield of gluconic acid. Dai et al showed low pH condition is suitable for gluconic acid production, because oxidation of gluconic acid is largely inhibited under low pH condition. The authors also showed this low pH strategy is applicable to gluconic acid production from lignocellulose hydrolysate. The findings on the GA production at low pH with no neutralizer added are some interesting, thus, I would like to recommend the manuscript to be publishable with a minor revision, such as the following modifications and questions to be addressed:
1. Line 22-23, Sentences of “Therefore, a low pH-stress strategy for efficiently producing gluconic acid was put forward and applied in gluconic acid production from biomass-derived glucose” should be simplified.
2. The initial enzymatic hydrolysate concentration should be mentioned in the section of Abstract.
3. low pH should be added as keywords.
4. Line 45-46, are there any chemical methods for producing GA?
5. Line 55-56, the sentences should be rewritten for better understanding.
6. Line 116, the glucose content after enzymatic hydrolysis?
7. How much CaCO3 was added?
8. Line 150, If the glucose if in the monosaccharide form (already released from cellulose), what is the source of the conversion factor "1.089" in the formula?
9. I think Gluconobacter spp. have the enzyme oxidizing gluconic acid to yield 5-ketogluconic acid. My concern is why the authors do not mention this enzyme. If the Gluconobacter strain used in this study does not have this enzyme, the authors should mention that. Otherwise, they should determine 5-ketogluconic acid in this work.
Author Response
1. Line 22-23, Sentences of “Therefore, a low pH-stress strategy for efficiently producing gluconic acid was put forward and applied in gluconic acid production from biomass-derived glucose” should be simplified.
Answer: Thanks for reviewer’s suggestion, we have revised the sentences as “Therefore, a low pH-stress strategy for efficiently producing gluconic acid from biomass-derived glucose was put forward.”
2. The initial enzymatic hydrolysate concentration should be mentioned in the section of Abstract.
Answer: As your suggestion, the initial glucose content was added in the sentence: “ 98.8 g/L gluconic acid with a yield of 96% could be obtained from concentrated corncob enzymatic hydrolysate that initially contained 100 g/L glucose…”
3. low pH should be added as keywords.
Answer: The low-pH as keyword was added.
4. Line 45-46, are there any chemical methods for producing GA?
Answer: GA could be chemically produced by oxidation of glucose with hypochlorite solution or in presence of bromide. Although, chemical conversion involves a single step, its limited selectivity towards GA, environmental toxicity, and biological hazards make it unfavorable, the bio-process can reduce side reactions during chemical production, making it an economically and attractive method for GA production. We have already revised the sentences.
5. Line 55-56, the sentences should be rewritten for better understanding.
Answer: The sentence was revised as “Gluconobacter oxydans (G. oxydans) has also been reported to produce GA and it catalyzes glucose to GA uses a different enzyme from Aspergillus Niger; G. oxydans was regarded as the preferable microorganism due to the better tolerance for lignocellulose degraded inhibitors and high content of sugars”
6. Line 116, the glucose content after enzymatic hydrolysis?
Answer: Sorry for the unclear description the glucose content after enzymatic hydrolysis with 10% solids loading was 56.5 g/L.
7. How much CaCO3 was added?
Answer: We are sorry for the unclear description. Here, 2 g CaCO3 (40g/L) was pre-loaded into the medium. We have already re-wrote the sentences.
8. Line 150, If the glucose if in the monosaccharide form (already released from cellulose), what is the source of the conversion factor "1.089" in the formula?
Answer: The molar mass of glucose and GA are 180 and 196, respectively. In the case of the glucose is completely bio-oxidized into GA, approximately 1.089 g GA is generated from 1 g glucose. Thus, the calculation of GA yield should divide by the conversion factor "1.089".
9. I think Gluconobacter spp. have the enzyme oxidizing gluconic acid to yield 5-ketogluconic acid. My concern is why the authors do not mention this enzyme. If the Gluconobacter strain used in this study does not have this enzyme, the authors should mention that. Otherwise, they should determine 5-ketogluconic acid in this work.
Answer: Thanks for reviewer’s carefulness and sorry for our negligence to description. G. oxydans has been well reported that gluconate is oxidized to 5-ketoglyconate by 5-gluconate dehydrogenase, and the corresponding reference is “C Prust, M Hoffmeister, H Liesegang, A Wiezer, WF Fricke, A Ehrenreich, G Gottschalk, U Deppenmeier, 2005. Complete genome sequence of the acetic acid bacterium Gluconobacter oxydans. Nature biotechnology, 23 (2005), 195-200”. The strain used in this study also had 5-gluconate dehydrogenase, and also produced 5-ketogluconate (5-KGA) at a later stage of fermentation, but the content was very low and it took a long time. Just little 5-KGA was determined (lower than 1.5 g/L) within 36 h fermentation. Based on the above considerations, we did not include 5-KGA in the study.
Reviewer 2 Report
This manuscript reported GA production from enzymatic hydrolysate of acid-treated corncob with low pH stress. It found that low pH could obtain purer GA and the GA production would not increase with the increase of glucose concentration. It would be useful for guiding the actual GA fermentation. The following questions should be solved.
1. The significance of this study should be described in the Introduction section.
2. Eq 3. 100% should be 100. How to calculate the yield of 2-KGA?
3. Since it mentioned high cost of current biorefinery process of lignocellulose, what about this process due to the vacuum concentration process and pH adjustment after enzymatic hydrolysis. The glucose concentration should be given after enzymatic hydrolysis of acid-treated corncob.
4. Uncomplete sentence in line 195.
5. English writing should be improved for the whole manuscript.
Author Response
1. The significance of this study should be described in the Introduction section.
Answer: Thanks for reviewer’s reminder, in this study, low pH or no neutralizer addition strategy the GA further oxidization. Eventually, a high yield of GA was achieved from lignocellulosic hydrolysate by controlling the end point of the catalysis with G. oxydans and this strategy contributes to subsequent isolation and purification. We have revised some sentences as your suggestion.
2. Eq 3. 100% should be 100. How to calculate the yield of 2-KGA?
Answer: As your suggestion, we have corrected it and the equation for 2-KGA yield calculation was added.
3. Since it mentioned high cost of current biorefinery process of lignocellulose, what about this process due to the vacuum concentration process and pH adjustment after enzymatic hydrolysis. The glucose concentration should be given after enzymatic hydrolysis of acid-treated corncob.
Answer: Sorry for the unclear description the glucose content after enzymatic hydrolysis with 10% solids loading was 56.5 g/L. The initial pH after diluted or concentrated was adjusted to 6.5 by adding NaOH. We have revised the Section 2.2.
4. Uncomplete sentence in line 195.
Answer: Thanks for reviewer’s carefulness and the sentence was revised as “It seemed that pH control was not necessary for the GA fermentation at this glucose concentration and no neutralizer addition contributes to subsequent isolation and purification.”
5. English writing should be improved for the whole manuscript.
Answer: We would like to thank the reviewer for the thoughtful review. English was improved by a professional language editor.
Reviewer 3 Report
In this study, a low pH-stress strategy for efficiently producing gluconic acid was put forward and applied in gluconic acid production from biomass-derived glucose using Gluconobacter oxydans. However, I think this manuscript only does research related to pH optimization, and the innovation and experimental content are insufficient for publication.
Figures 1 and 2 are too informative to understand. I suggest that Figure 1 can be split into two figures. In Figure 2, do not mix the line chart and the bar chart. Some data can be put into supplementary materials.
Author Response
1. In this study, a low pH-stress strategy for efficiently producing gluconic acid was put forward and applied in gluconic acid production from biomass-derived glucose using Gluconobacter oxydans. However, I think this manuscript only does research related to pH optimization, and the innovation and experimental content are insufficient for publication.
Answer: In this manuscript, the growth of Gluconobacter oxydans and the performance of fermentation in different pH conditions were investigated and the low pH-stress strategy (without any neutralizers addition) was found to be an effective method for improving the yield of gluconic acid. Based on the pH experiments, the gluconic acid production was also tested using enzymatic hydrolysate from corncob. As a result, 98.8 g/L gluconic acid with a yield of 96% could be obtained from concentrated corncob enzymatic hydrolysate that initially contained 100 g/L glucose with 1.5 g/L cells loading of G. oxydans. The results verify that low pH stress could effectively control end-point, which provide a potential strategy for industrial gluconic acid production from lignocellulosic materials. Thus, we would be grateful if the manuscript could be considered for publication in this journal.
2. Figures 1 and 2 are too informative to understand. I suggest that Figure 1 can be split into two figures. In Figure 2, do not mix the line chart and the bar chart. Some data can be put into supplementary materials.
Answer: Thanks for reviewer’s suggestion, the Figure 1 and 2 were split.

Round 2
Reviewer 2 Report
Although most questions are solved, the following questions should be treated seriously.
1. It should not mention the economic issue in the Introduction section due to no economic analysis in this work.
2. Many syntax errors appeared in the text, please check the whole manuscript carefully. Some errors were selected as follow.
Line 46, “… and it exhibit…” should be “… and it exhibits…”
Lines 47-48, “…due to the purity… is not high,…” has syntax error.
Lines 50-51, this sentence should be deleted due to no economic analysis.
Line 55, “…it limited selectivity…” should be “…its limited selectivity…”
Line 62, “Comparisons shows…” should be “Comparisons show…”
Line 70,”and it catalyzes glucose to GA uses…” has syntax error.
Author Response
Thanks for reviewer’s suggestion and carefulness, the content about economic issue was deleted and these errors your mentioned were corrected. In addition, we have double checked the manuscript and revised some other errors.
Reviewer 3 Report
I don't have any suggestions. I suggest accepting it
Author Response
Thanks for your accept.